# Widespread conservation and lineage-specific diversification of genome-wide DNA methylation patterns across arthropods

**Samuel H. Lewis**[1,2,3], **Laura Ross**[4], **Stevie A. Bain**[4], **Eleni Pahita**[2,3], **Stephen A. Smith**[5], **Richard Cordaux**[6], **Eric A. Miska**[1,7], **Boris Lenhard**[2,3,8], **Francis M. Jiggins**[1☯]*, **Peter Sarkies**[2,3☯]*

**1** Department of Genetics, University of Cambridge, Cambridge, United Kingdom, **2** MRC London Institute of Medical Sciences, London, United Kingdom, **3** Institute of Clinical Sciences, Imperial College London, London, United Kingdom, **4** Institute of Evolutionary Biology, Edinburgh, United Kingdom, **5** Department of Biomedical Sciences and Pathobiology, Virginia Maryland College of Veterinary Medicine, Virginia Tech, Blacksburg, Virginia, United States of America, **6** Laboratoire Ecologie et Biologie des Interactions Universite de Poitiers, France, **7** Wellcome Trust/Cancer Research UK Gurdon Institute, Cambridge, United Kingdom, **8** Sars International Centre for Marine Molecular Biology, University of Bergen, Bergen, Norway

☯ These authors contributed equally to this work.

* fmj1001@cam.ac.uk (FMJ); psarkies@imperial.ac.uk (PS)

**Data Availability Statement:** Sequence data that was newly-generated for this project have been deposited in the NCBI Short Read Archive under the BioProject accession code PRJNA589724. The

## Abstract

Cytosine methylation is an ancient epigenetic modification yet its function and extent within genomes is highly variable across eukaryotes. In mammals, methylation controls transposable elements and regulates the promoters of genes. In insects, DNA methylation is generally restricted to a small subset of transcribed genes, with both intergenic regions and transposable elements (TEs) depleted of methylation. The evolutionary origin and the function of these methylation patterns are poorly understood. Here we characterise the evolution of DNA methylation across the arthropod phylum. While the common ancestor of the arthropods had low levels of TE methylation and did not methylate promoters, both of these functions have evolved independently in centipedes and mealybugs. In contrast, methylation of the exons of a subset of transcribed genes is ancestral and widely conserved across the phylum, but has been lost in specific lineages. A similar set of genes is methylated in all species that retained exon-enriched methylation. We show that these genes have characteristic patterns of expression correlating to broad transcription initiation sites and well-positioned nucleosomes, providing new insights into potential mechanisms driving methylation patterns over hundreds of millions of years.

## Author summary

Animals develop from a single cell to form a complex organism with many specialised cells. Almost all of the fantastic variety of cells must have the same sequence of DNA, and yet they have distinct identities that are preserved even when they divide. This remarkable process is achieved by turning different sets of genes on or off in different types of cell using molecular mechanisms known as "epigenetic gene regulation". Surprisingly, though

source code, input data and newly-identified DNMT & ALKB2 gene sequences are available from the Cambridge Data Repository (https://doi.org/10.17863/CAM.45964). Bisulfite sequencing data for Planococcus citri has been deposited PRJNA610765.

**Funding:** This research was funded by the Medical Research Council (grant to PS) and the Leverhulme Trust (grant to FMJ and PS).BL was supported by Medical Research Council UK (award MC_UP_1102/1) and Wellcome Trust Investigator Award 106954/Z/15/Z. LR was supported by a NERC fellowship (NE/K009516/1) and the Royal Society (RG160842). The funders had no role in study design, data collection and analysis, decision to publish or preparation of the manuscript.

**Competing interests:** The authors have declared that no competing interests exist.

all animals need epigenetic gene regulation, there is a huge diversity in the mechanisms that they use. Characterising and explaining this diversity is crucial in understanding the functions of epigenetic pathways, many of which have key roles in human disease. We studied how an epigenetic regulation known as DNA methylation has evolved within arthropods. Arthropods are an extraordinarily diverse group of animals ranging from horseshoe crabs to fruit flies. We discovered that the levels of DNA methylation and where it is found within the genome changes rapidly throughout arthropod evolution. Nevertheless, there are some features of DNA methylation that seem to be the same across most arthropods- in particular we found that there is a tendency for a similar set of genes to acquire methylation of DNA in most arthropods, and that this is conserved over 350 million years. We discovered that these genes have distinct features that might explain how methylation gets targeted. Our work provides important new insights into the evolution of DNA methylation and gives new hints to its essential functions.

## Introduction

In most organisms DNA bases are adorned with a variety of chemical modifications. Amongst the most common of these is methylation at the 5 position of cytosine (C5me), which is present from bacteria to humans [1–3]. In eukaryotes, a key property of cytosine DNA methylation is its ability to act epigenetically—that is, once introduced, methylation at specific cytosines can remain in place through cell division [4],[5]. This relies on the activity of "maintenance" methyltransferases, DNMT1 in animals [6], which recognise CG dinucleotides (CpG sites) where one strand is methylated and one strand unmethylated and catalyse the introduction of methylation on the unmethylated strand [4]. Meanwhile "de novo" methyltransferases act on unmethylated DNA. In animals this role is performed by DNMT3 enzymes, which introduce 5meC predominantly within CpG sites [4]. Mechanisms also exist to remove methylation from DNA, including the TET family of enzymes, which convert 5meC to a hydroxymethylated intermediate which can be removed by base excision repair or diluted out through cell division [5]. As the maintenance and de novo methylation of CG sequences occurs through the activity of homologous enzymes in plants and animals [6], CpG methylation was likely present among the earliest eukaryotic organisms.

In mammals, a key function of CpG methylation is to defend the genome against transposable elements (TEs) by preventing their transcription and transposition [7], and loss of DNA methylation leads to reactivation of TEs [8]. CpG methylation targeted to the promoters of genes can also suppress transcription, typically resulting in stable silencing [7]. Another notable genome-wide pattern is the enrichment of CpG methylation within the exons of transcribed genes [9]. In contrast to TE and promoter methylation, this is not associated with transcriptional silencing.

Whilst CpG methylation at both TEs and gene bodies is present in both plants and animals [3], across eukaryotic species both DNA methylation levels and the targets of methylation have evolved rapidly [10,11]. Most strikingly, CpG methylation has been independently lost altogether several times, coinciding with the loss of DNMT1 and DNMT3 DNA methyltransferase enzymes [6,10,11]. Across eukaryotes, loss of CpG methylation tends to be accompanied by loss of the DNA alkylation repair enzyme ALKB2, which repairs damage caused by DNMTs introducing 3-methylcytosine into DNA. This suggests that some species correct DNA alkylation using ALKB2, and others avoid it altogether by losing the DNA methylation pathway [12]. Even within species that retain DNA methyltransferases, the genomic distribution of

CpG methylation differs widely [10–16]. Such variability is surprising given the essential role of CpG methylation in genome regulation in mammals and plants, and there are few clues as to what factors drive the changes. Tracing the evolution of CpG methylation is currently challenging because detailed descriptions of DNA methylation patterns are patchy and focussed on model systems, leaving large parts of the phylogenetic tree underexplored.

Here we study CpG methylation patterns across arthropods. Arthropods have been suggested to represent a very different system of CpG methylation from mammals [17]. Whilst the well-characterised model organism *Drosophila melanogaster* lacks DNA methylation altogether, DNA methyltransferases 1 and 3 were found in the honey bee *Apis mellifera* [18]. Genome-wide CpG methylation mapping demonstrated that methylation was globally extremely low, and restricted to the bodies of a subset of transcribed genes [10,19]. Subsequently, similarly restricted patterns of DNA methylation were found in other insects [19–22]. Such patterns support the proposal that gene body methylation is conserved across eukaryotes while TE methylation has been lost altogether in arthropods [10,17]. However some insects show considerably higher levels of genome-wide methylation [13], and variation in arthropod methylation levels also exists outside of insects [15,23–26]. There is also evidence of TE methylation in the desert locust *Schistocerca gregaria* [19]. A thorough reconstruction of the evolution of methylation across the phylum is still lacking.

We set out to explore the evolution of arthropod methylation patterns by characterising genome-wide CpG methylation across the phylum. We show that TE methylation was ancestral to arthropods, although at a relatively low level. Methylation of protein-coding genes was also ancestral to arthropods, with similar subsets of genes being targeted for methylation across arthropods. Despite these conserved features, we find many examples of diversification in methylation patterns across arthropods, in particular loss of gene methylation in crustaceans and gain of both promoter methylation and genome-wide TE methylation in the myriapod *Strigamia maritima* and the insect *Planococcus citri*. We find that methylation at genes, enriched within exons, is the most widely conserved feature of arthropod methylomes and we use comparative analysis to identify a link between exon methylation and nucleosome positioning. Overall, our findings demonstrate that while key features of global methylation patterns have been conserved across millions of years of arthropod evolution, the targets of DNA methylation can rapidly diverge within individual lineages.

## Results

### Genome-wide levels of CpG methylation vary widely across the arthropods

We carried out high-coverage whole-genome bisulphite sequencing (WGBS) on 14 species of arthropod and quantified the levels of DNA methylation with base-pair resolution. To examine genome-wide methylation levels we combined this data with published results from 15 additional species [13,24,27,28]. Estimates of genome-wide CpG methylation were then used to reconstruct ancestral methylation levels across the arthropod phylogeny. All 18 species of holometabolous insects had low levels of CpG methylation, and this was likely the ancestral state of this clade (Fig 1A and 1B). While CpG methylation rates in other arthropod clades tended to be higher, they varied considerably (Fig 1A and 1B). The ancestral arthropod likely had moderate methylation levels (9.1±4.8%; Fig 1A) but higher methylation levels evolved in *S. maritima*. Similarly, the ancestor of insects had methylation levels lower than some taxa such as *B. germanica* (4.0±3.2% versus 10.9%) indicating that methylation level fluctuated throughout arthropod evolution.

To investigate the evolution of the DNA methylation machinery across arthropods, we searched the genomes of these species for homologues of the genes encoding the

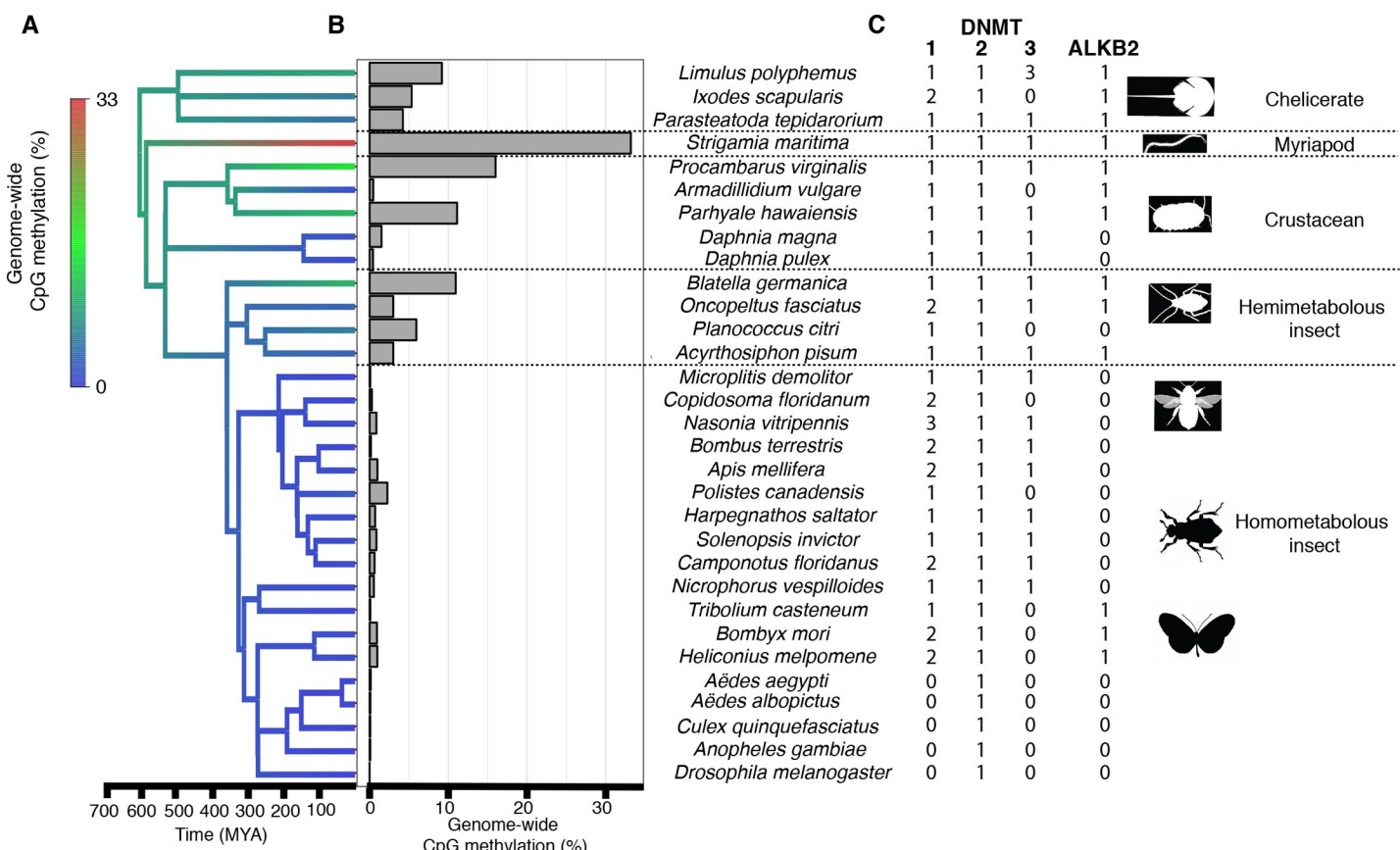

**Fig 1. Genome-wide CpG methylation across the arthropod phylogeny.** (A) A phylogeny of 29 31 arthropod species that have publicly available or newly computed genome-wide methylation estimates, with branches coloured to show an ancestral state reconstruction of the percentage of CpG sites that are methylated in the genome. (B) The percentage of CpG sites that are methylated genome-wide. (C) The number of *DNMT* and *ALKB2* homologues in the genomes of each species.

methyltransferases DNMT1-3. We confirmed the genes all encoded a full cytosine methyltransferase domain, and where we did not find annotated homologues we directly search the genomic DNA for unannotated genes. In each species we found a single copy of *DNMT2*, which methylates tRNAs [29] (Fig 1C). *DNMT1* was present in all species apart from the five Diptera (Fig 1C). The loss of this gene was associated with the loss of CpG methylation (Fig 1C), with methylation rates in *D. melanogaster* not significantly different from the unmethylated DNA spike-in included in each reaction. *DNMT3* was absent from the genomes of 14 species, with inspection of the tree suggesting at least eight independent losses (Fig 1C). Several of these species possessed moderate levels of CpG methylation (Fig 1B), indicating that DNMT1 alone can be sufficient for introducing genome-wide DNA methylation, consistent with earlier studies in arthropods and nematodes [12,13,20].

Across the eukaryotes ALKB2, which repairs DNA alkylation damage introduced by DNMTs, tends to be lost from the same taxa as DNMT1 and 3 [12]. Arthropods exhibited many exceptions to this general rule—there have been at least five losses of ALKB2 but only one of these is associated with the loss of DNMT1 and 3 (Fig 1C). However, we found that species with ALKB2 possessed higher methylation levels (S1 Fig.; phylogenetic mixed model: $p = 0.0194$), suggesting ALKB2 is dispensable in species with low levels of DNA methylation.

## Rapid loss and gain of TE methylation across arthropods

In mammals, plants and nematodes, transposable elements (TEs) are preeminent targets of DNA methylation, but previous studies have shown that TE methylation is rare in holometabolous insects [10,11,19,21,22]. However, DNA methylation has been found at TEs in some arthropods [15,23,25,30]. To explore the distribution of TE methylation across arthropods we annotated transposable elements using RepeatMasker analysis of the entire genome, and removed annotations that did not contain Pfam domains derived from transposable elements. We focused on 14 species that represent the diversity of arthropods, and have assembled and annotated genomes (see Fig 2B).

Compared to unannotated regions of the genome, TEs were strongly enriched for methylation in *S. maritima* and *P. citri*, and weakly enriched in several other species (Fig 2B and 2C). This pattern is reflected in the distribution of methylation across TEs—this is skewed towards 0% for most species, but in *S. maritima* and *P. citri* the large majority of TEs are methylated (Fig 2A,B; S2 Fig). In these two species there was a sharp drop in methylation rates at the boundary of the TE (Fig 2D). In agreement with earlier studies [19,22], the methylation rate of TEs was low in holometabolous insects. However, outside of this group there was moderate methylation of TEs in chelicerates (*L. polyphemus* and *P. tepidariorum*), the crustacean *P. hawaiensis* and hemimetabolous insects (*B. germanica* and *A. pisum*) (Fig 1A and 1C). To further quantify the extent of TE methylation, we clustered TEs into highly- and lowly-methylated groups in each species separately, and calculated the proportion of TEs that were assigned to the highly-methylated group (Table 1). Ancestral state reconstruction suggested that a low level of TE methylation was present in the ancestral arthropod, but was lost in the ancestor of holometabolous insects (Fig 2A). Contrastingly, the large majority of TEs were targeted by methylation in *S. maritima* and *P. citri*, suggesting independent acquisition of TE methylation in these species.

To investigate changes to the DNA methylation machinery that accompanied the gain of TE methylation, we examined the DNMT genes found in *S. maritima* and *P. citri*. We did not find *DNMT3* homologues in the genome of *P. citri*. However, both species contain *DNMT1* genes that have lost the CXXC domain, despite this domain being conserved in 8 species that lacked TE methylation (S3 Fig and S4 Fig; incomplete genome assembly prevented some species from being examined). This domain has been suggested to contribute to the specificity of DNMT1 for hemimethylated DNA [31], suggesting that DNMT1 may act on unmethylated DNA in these species. It is possible that independent loss of this domain in *DNMT1* homologues in the two species might be associated with acquisition of TE methylation. Alternatively, in the fungus *Cryptococcus neoformans* it has been suggested that high levels of methylation of features including TEs reflect long-term maintenance of DNA methylation coupled with stochastic changes to methylation and selection on beneficial methylated variants [32].

## Methylation at exons is conserved across most arthropods

We next investigated methylation at genes across arthropods. In all but one of the species we tested, mean methylation levels across exons were significantly higher than unannotated regions of the genome (Fig 3B). The exception was *P. hawaiensis*, where exons are significantly less methylated than unannotated regions of the genome (Fig 3B). There is a significant difference between methylation at exons and introns in *P. hawaiensis* (p = 0.001, paired t test). In the species with exon methylation, the distribution of methylation suggested that a subset of genes is targeted for methylation (Fig 3C). When clustered into highly and lowly methylated genes, the proportion of methylated genes varied similarly to mean methylation across genes (Table 1).

To investigate the distribution of methylation within genes, we compared the methylation levels at exons and introns in each species. Methylation was higher at exons in the majority of

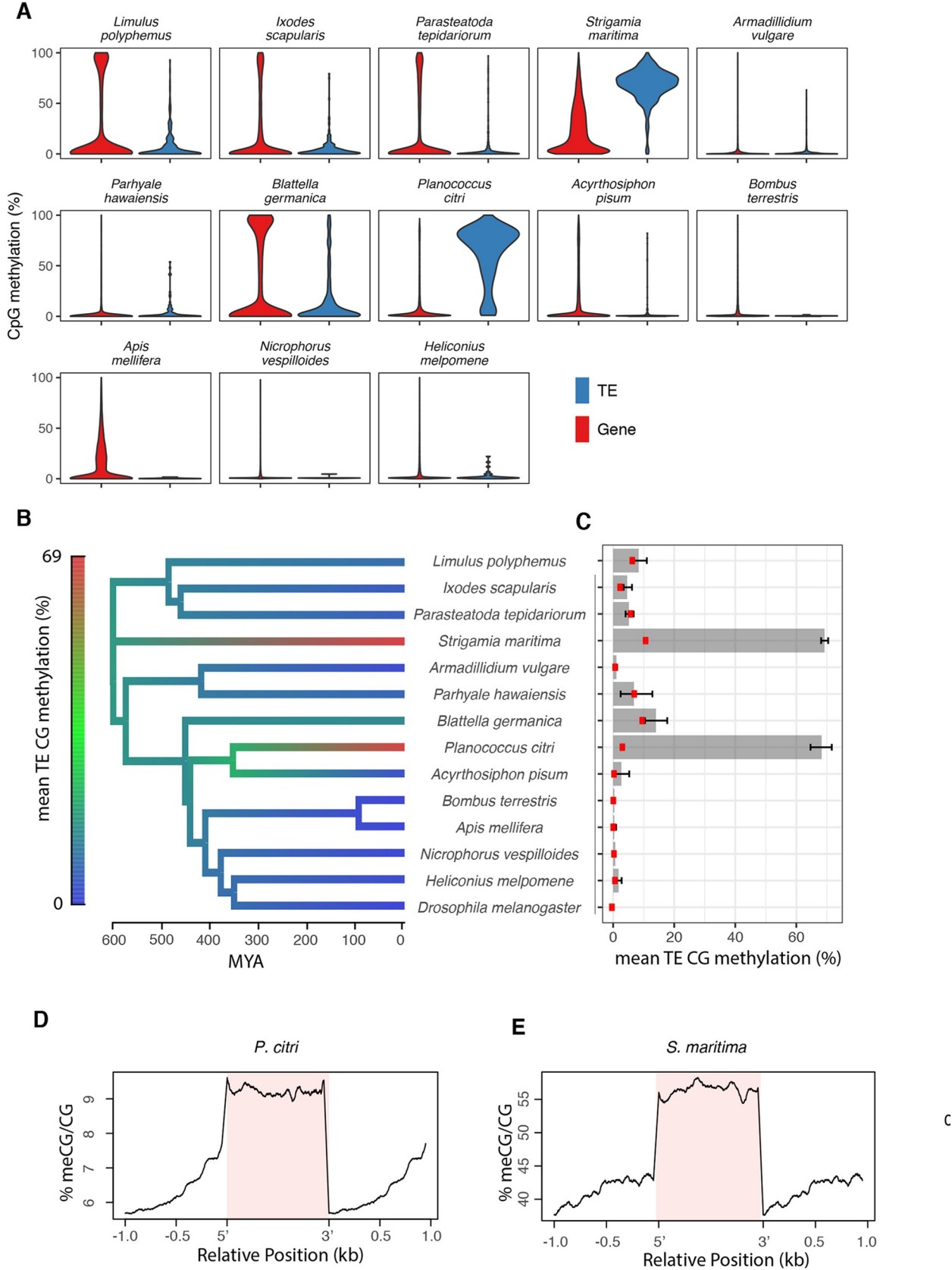

**Fig 2. Methylation of transposable elements.** For 14 diverse arthropod species with annotated genomes, we explored methylation characteristics of genomic features. (A) Density plot of the mean % CpG methylation per gene and per TE. (B) Ancestral state reconstruction of the mean % methylation of CpGs within TEs. (C) Mean % methylation of CpGs within TEs with 95% bootstrap confidence intervals. Red points are CpGs >1kB from annotated regions of the genome. (D,E) Metagene plot of methylation within TEs (pink) and in flanking sequence for *S. maritima* and *P. citri*.

species, suggesting that the gene body methylation in arthropods is due to targeting of methylation to exons. However, there was little difference between exons and introns for the two crustaceans, *P. hawaiensis* and *A. vulgare* (Fig 3C; S3 Fig). Methylation of genes has been described in the crayfish [26] and suggested in *Daphnia* [24] albeit at very low levels. Given that *P. hawaiensis* exons are depleted for methylation relative to the genome-wide background while *A. vulgare* exons are only slightly greater than the background, this may reflect an ancient loss of gene body methylation in the Peracaridian ancestor of these species. Among species with exon methylation, there were differences in how methylation levels changed across the gene (Fig 3C). For example, methylation was largely confined to the first three exons of *P. citri* and *N. vespilloides*, while methylation in *B. germanica* is largely found from exon four onwards (Fig 3C). However, there were no clear phylogenetic trends within these patterns suggesting patterns of methylation across genes likely change frequently during evolution. Together these data suggest that exon-enriched methylation was an ancestral property of arthropod methylomes which is largely conserved across the phylum.

## Independent acquisition of promoter methylation in arthropod lineages

In mammals, methylation of regions immediately upstream of genes, often at CpG islands, is associated with gene silencing. However, there is no evidence of promoter methylation in insects [19,20,22]. To examine promoter methylation associated with gene silencing across arthropods, we extracted 1kb upstream of genes for all species. In most species there was little difference in upstream methylation between high and low expression genes; however, low expression genes in *P. citri* and *S. maritima* had significantly higher upstream methylation

**Table 1. Proportion of Genes and TEs that are highly methylated.**

| Species | TEs[a] | | Genes | |
|---|---|---|---|---|
| | Number | Proportion methylated[b] | Number | Proportion methylated[b] |
| *Acyrthosiphon pisum* | 293 | 0.017 | 13147 | 0.171 |
| *Apis mellifera* | 7 | 0.143 | 10066 | 0.272 |
| *Armadillidium vulgare* | 655 | 0.020 | 4703 | 0.019 |
| *Blattella germanica* | 276 | 0.145 | 9272 | 0.387 |
| *Bombus terrestris* | 78 | 0.128 | 8550 | 0.069 |
| *Heliconius melpomene* | 34 | 0.088 | 11583 | 0.077 |
| *Ixodes scapularis* | 212 | 0.033 | 5775 | 0.219 |
| *Limulus polyphemus* | 342 | 0.117 | 7227 | 0.265 |
| *Nicrophorus vespilloides* | 9 | 0.111 | 12305 | 0.032 |
| *Parasteatoda tepidariorum* | 622 | 0.032 | 9742 | 0.243 |
| *Parhyale hawaiensis* | 89 | 0.079 | 3302 | 0.028 |
| *Planococcus citri* | 361 | 0.751 | 34044 | 0.099 |
| *Strigamia maritima* | 719 | 0.758 | 12898 | 0.326 |

[a] TEs with annotated TE-associated domains (see Methods);

[b] the proportion falling into the highly methylated group after clustering each feature type within each species

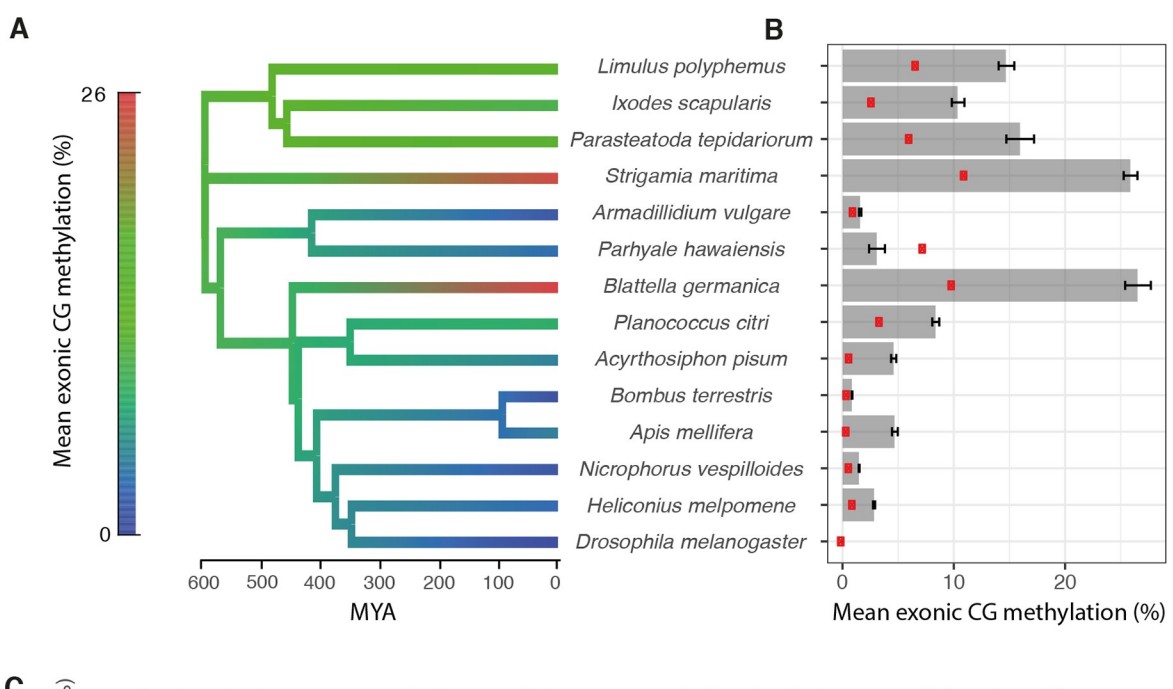

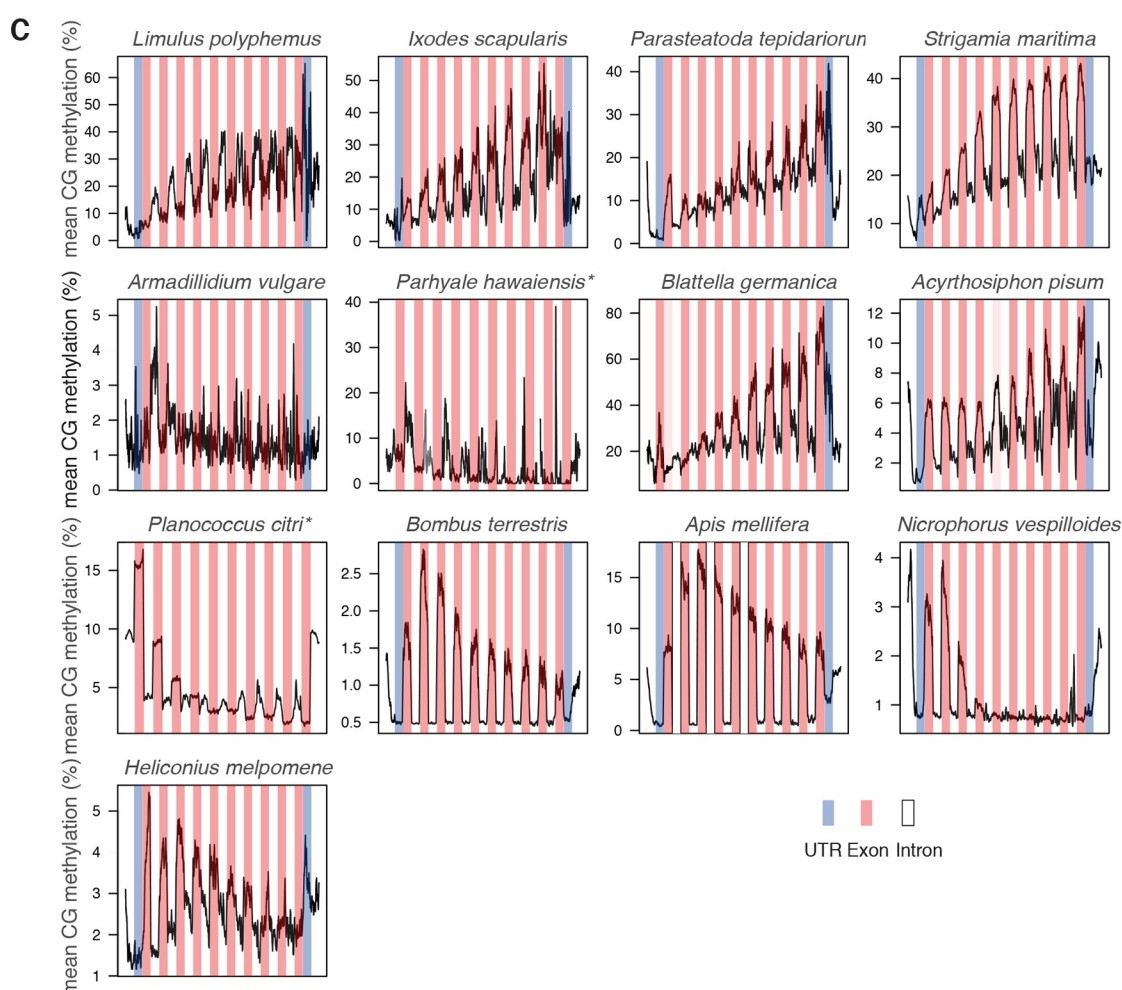

**Fig 3. Gene body methylation.** (A) Ancestral state reconstruction of the mean % methylation of CpGs within exons. (B) Mean % methylation of CpGs within exons with 95% bootstrap confidence intervals. Red points are CpGs >1kB from annotated regions of the genome. (C) Metagene plot of methylation across introns (white), exons (pink), UTRs (blue) and 1kB of flanking sequence (white).

than high expression genes (Fig 4A). In *S. maritima* only genes with very high upstream methylation showed clearly reduced gene expression (p = 1e-15, Kruskal Wallis test), whilst in *P. citri* there was a positive correlation between upstream methylation and gene expression across a wider range of upstream methylation levels (Fig 4B). The different relationship between upstream methylation and gene expression between *S. maritima* and *P. citri* and the lack of a similar relationship in other arthropod species suggests that promoter methylation associated with gene silencing may have evolved independently in these two species.

## Methylated genes are conserved and have moderate to high expression

Our results suggest that the most highly conserved feature of arthropod methylomes is enrichment of methylation at the exons of a subset of genes. Similar conclusions as to the dominance of methylation within genes in arthropods have been reached through a systematic analysis of insects [13] and in analyses of several individual arthropod species outside insects [24–26,30]. Across species, we asked whether there was any tendency for orthologous genes to be methylated in different species. We ranked orthologous genes by relative methylation levels across species and observed that there was a clear tendency for orthologs to have similar levels of methylation in different species (Fig 5A). The observation that the same genes are methylated in different species raised the question of what determines which genes acquire methylation. We used comparative analysis to investigate this across the phylum.

Methylation has been shown to be enriched at alternatively spliced genes in some insects [19,22]. To test for a link between methylation and splicing across arthropods, we compared the level of methylation between genes with one exon (which cannot undergo splicing) and genes with two or more exons (which may undergo splicing). We found no clear difference in any species (S5 Fig), suggesting that splicing does not explain the propensity of genes to acquire methylation across arthropods.

Previously, methylation of genes in individual insect species has been correlated to higher levels of expression [20,22]. We find a statistically significant tendency for genes with high methylation to have higher expression across most species. However, many highly expressed genes are not methylated. Instead a more prominent trend is for methylated genes to have more focussed levels of gene expression such that genes with very low expression levels are rarely methylated (Fig 5B and 5C; S6 Fig). Curiously, this pattern is reversed in *P. citri*, where the exons of methylated genes tend to have low expression (S6 Fig).

Previously it has been noted that genes with high levels of methylation in arthropods are more likely to perform conserved "housekeeping" functions [26,33,34]. We clustered genes into orthologous groups across species and examined genes that were conserved across all species compared to species-specific genes. Across all species carrying gene body methylation, conserved genes with moderate to high expression were more likely to be methylated (Fig 5C; S6 Fig). Orthologues of genes with high methylation were strongly enriched for genes annotated as housekeeping genes in *D. melanogaster* on the basis of consistent expression across developmental timepoints and tissues [35] (50% of methylated genes had housekeeping functions as opposed to 15% of unmethylated genes; Fisher's Exact Test: $p = 2 \times 10^{-16}$; Fig 5D). Nevertheless many conserved and highly expressed genes, including those annotated as housekeeping genes, were not highly methylated, suggesting that neither conservation nor expression is sufficient to explain gene body methylation.

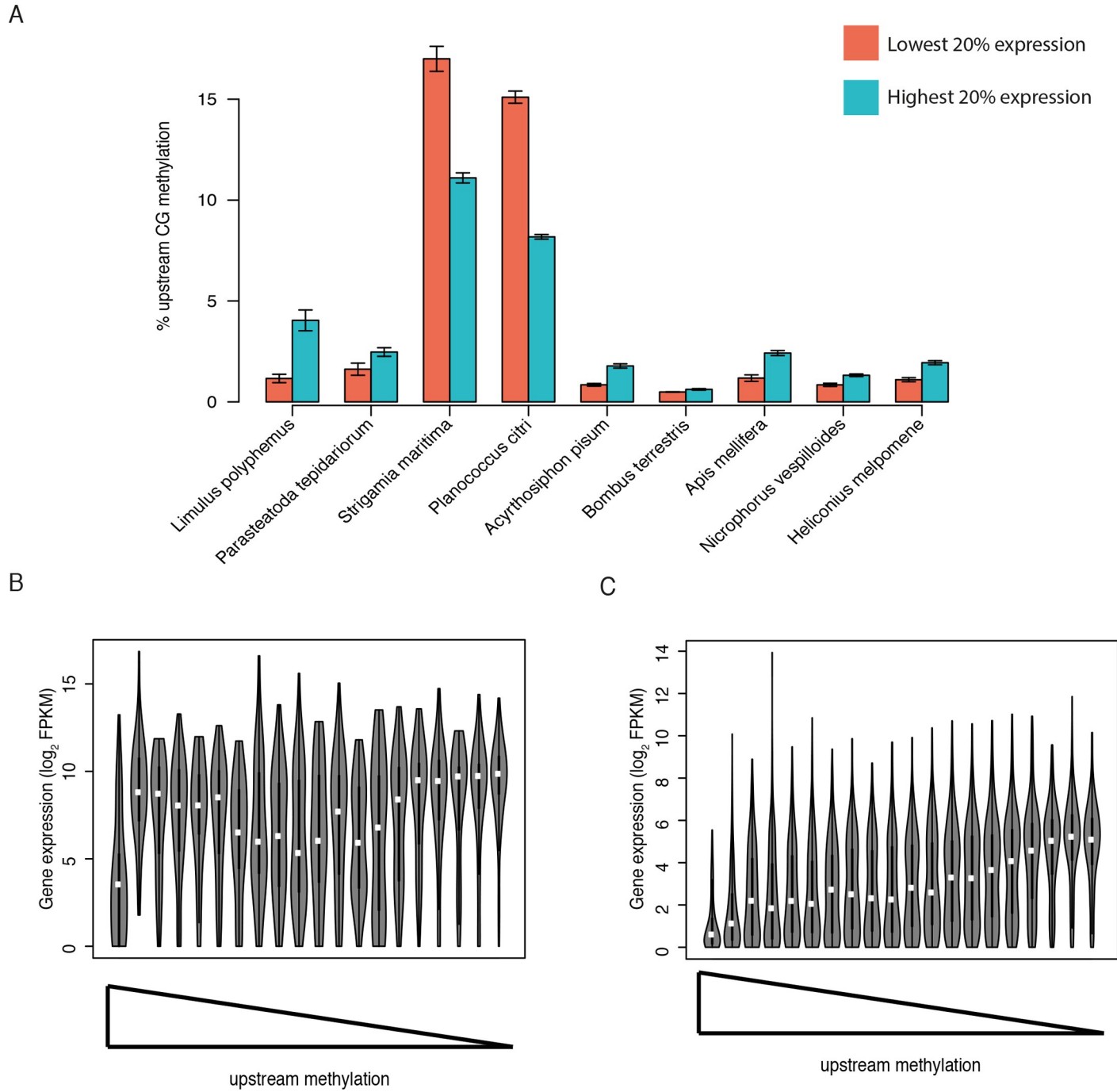

**Fig 4. Promoter methylation.** (A) Methylation across upstream regions for highly expressed genes (top 20%) and lowly expressed genes (bottom 20%). *P. hawaiensis* is omitted due to lack of gene expression data. Expression of genes across bins of decreasing upstream methylation in *S. maritima* (B) and *P. citri* (C).

## Nucleosome positioning influences DNA methylation levels across arthropods

In order to investigate molecular mechanisms that might be responsible for influencing DNA methylation we examined how the correlation in methylation between pairs of CpGs varied with increasing separation. In many species with exon-enriched methylation the correlation

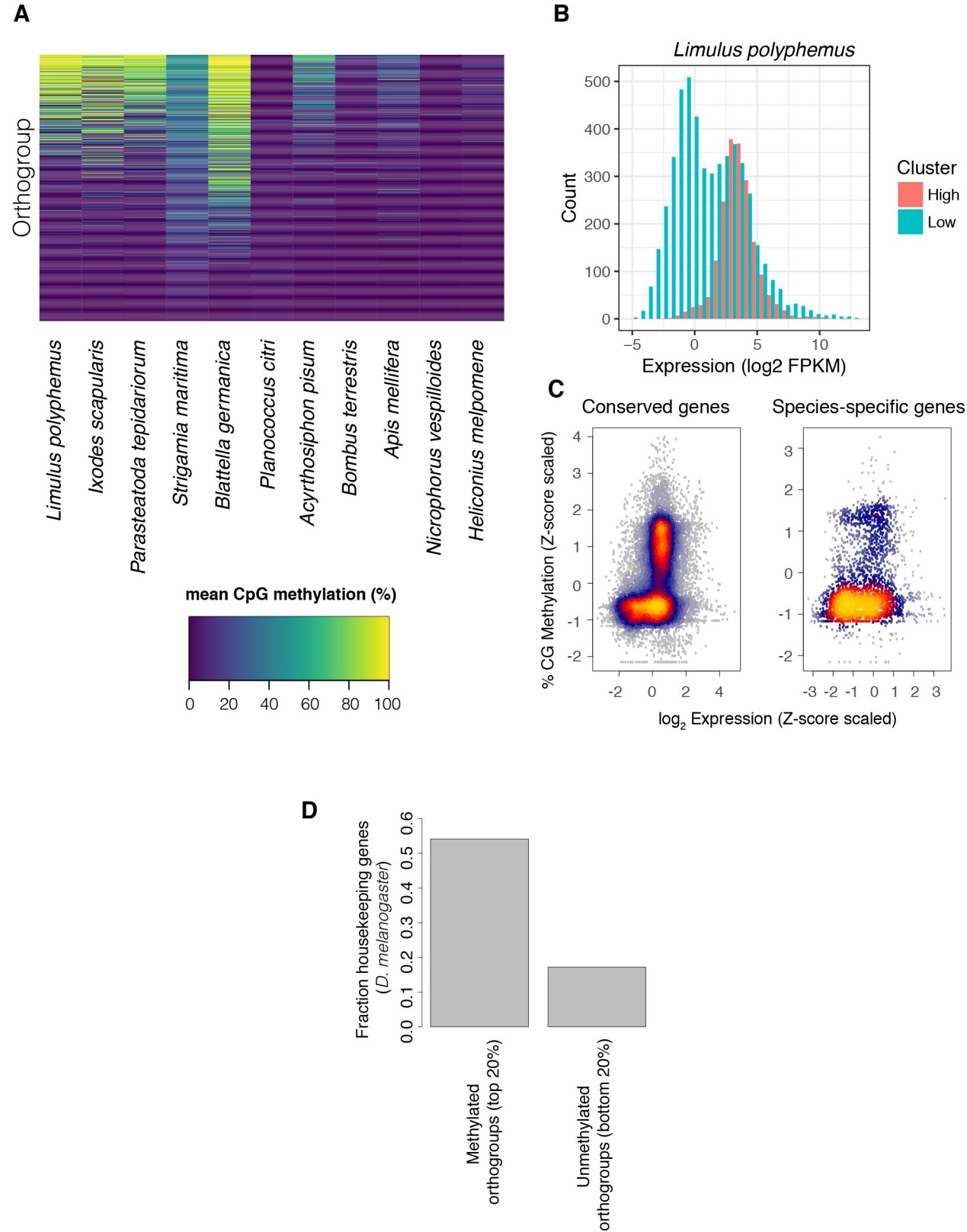

**Fig 5. The expression and conservation of methylated genes.** (A) Methylation of orthologous genes in different species. Only genes with orthologs in all species are shown, and in species with multiple paralogs the mean % CpG methylation is shown. Genes are ranked by their mean methylation.

(B) Histogram of gene expression estimated from RNAseq data for methylated and unmethylated genes in *L. polyphemous* (FPKM: fragments per kilobase million). (C) The relationship between gene expression and CpG methylation for genes that are conserved across all species and species-specific genes. To combine data across species, the methylation rate was normalised by taking the Z-score of methylation and expression of each gene within each species. Each point is a gene from a single species, and the colour represents the density of overlaid points. (D) Fraction of housekeeping genes as defined in references [37] for the top 20% and bottom 20% of methylated genes.

coefficient between methylation levels of individual CpGs oscillated periodically (Fig 6A and 6B). Fourier analysis showed that the period of oscillation was ~160 nucleotides, roughly corresponding to the average nucleosome repeat length (Fig 6A and 6B; S7 Fig). We quantified this nucleosome-length periodicity within exons across all species. While the majority of species with exonic methylation displayed a potential nucleosome-length periodicity signal, its magnitude varied greatly–for example *H. melpomene* has gene methylation but less apparent periodicity (Fig 6B). Interestingly a clear signal of periodicity was also seen for TE methylation in *S. maritima* and *P. citri*, both of which have high levels of TE methylation (S7 Fig).

We wondered whether the periodicity in correlation between methylated DNA might reflect an influence of nucleosome positioning on DNA methylation, as has been shown in plants [36] and inferred from analysis of mammalian DNA methylation profiles[37]. Such a connection has not yet been investigated in arthropods. We did not have genome-wide nucleosome positioning data for the majority of species so decided to investigate high-quality nucleosome positioning from *Drosophila* [38], examining orthologues of genes either enriched or depleted for DNA methylation across arthropods. The promoters of methylated genes possessed high nucleosome occupancy overall and strongly positioned nucleosomes just upstream (-1) and downstream (+1) of the transcription start site (TSS) (Fig 6C). The promoters of unmethylated genes showed lower nucleosome occupancy overall and demonstrated weaker positioning of the -1 and +1 nucleosome. Previous analyses of promoter types across eukaryotes have indicated that promoters with strong positioning of nucleosomes lead to initiation of transcription across a broad region (broad TSS) whilst promoters with weaker nucleosome positioning tend to have a much narrower TSS focussed around a dominant initiation site[39]. Using cap analysis of gene expression (CAGE) data from *D. melanogaster* we found that the TSS of *D. melanogaster* orthologs of methylated genes was broader than the TSS of orthologs unmethylated genes (Fig 6C).

Further evidence for a connection between nucleosome occupancy and a periodic signal in the correlation between methylation sites comes from a comparison of exons and introns. Exons are known to have much higher nucleosome occupancy than introns and accordingly the periodic signal of methylation correlation is markedly weaker in introns than in exons (S8 Fig). Together this supports a potential role for nucleosome occupancy in shaping CpG methylation patterns in arthropods.

The patterns of nucleosome occupancy and transcription initiation corresponded to previous analyses across organisms demonstrating that housekeeping genes tend to have well positioned nucleosomes just downstream of promoters and broad TSS whereas tissue-specific genes tend to have less well-defined nucleosome positions at promoters and narrow TSS [40–43]. We therefore tested whether methylated genes were more likely to have tissue-specific or global gene expression using RNAseq data from different tissue types. In every species with gene body methylation, we found that methylated genes tended to have less variable expression across different tissues (Fig 6D). Altogether this suggests that across arthropods conserved genes with strongly positioned nucleosomes, broad TSS and housekeeping functions are targeted for methylation whilst tissue-specific genes with opposite patterns of nucleosome occupancy and TSS width tend to be depleted of methylation.

Given that DNA methylation, nucleosome occupancy and constitutive expression are all correlated with each other, we wondered whether nucleosome occupancy influences DNA

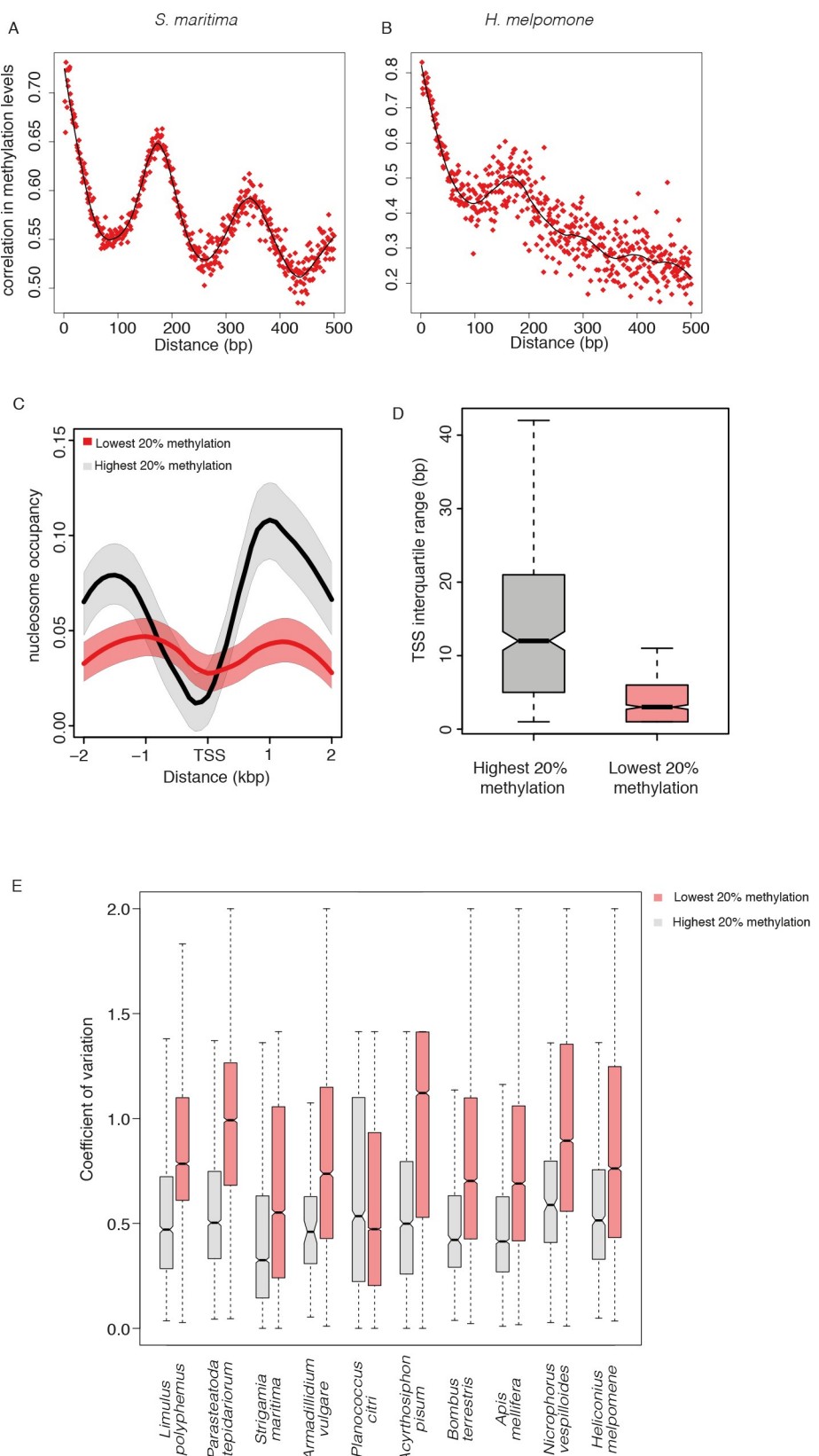

**Fig 6. Nucleosome occupancy and DNA methylation.** The Pearson correlation coefficient in DNA methylation levels between pairs of CpG at different distances apart in (A) *S. maritima* and (B) *H. melpomene*. (C) Nucleosome occupancy in *D. melanogaster* orthologues of genes that are either highly methylated (grey) or unmethylated (red) in arthropods. Shaded area is a 95% bootstrap confidence interval. Across all species in the dataset, mean methylation levels were estimated for each group of orthologous genes using a general linear mixed model. The top and bottom 20% were classified as methylated and unmethylated respectively. Only genes with orthologs in all species are shown. (D) Interquartile range of the TSS window for the *D. melanogaster* orthologues of highly methylated orthogroups (top 20%) and lowly methylated orthogroups (bottom 20%). (E) The coefficient of variation in expression of genes with high (top 20%) and low (bottom 20%) methylation across different tissues estimated using RNAseq data. *P. hawaiensis* is omitted because no tissue-specific data is available for this species.

methylation directly. To examine this, we compared nucleosome occupancy at the *Drosophila* orthologues of methylated genes that were annotated as housekeeping or not (S9 Fig). Importantly, the pattern of nucleosome positioning was similar for both methylated genes with housekeeping functions and methylated genes without housekeeping functions. Furthermore, housekeeping genes with low levels of methylation had low levels of nucleosome occupancy at promoters (S9 Fig). This implies that nucleosome positioning influences DNA methylation separately from its connection to the housekeeping functions. Therefore, patterns of nucleosome occupancy may explain why genes with housekeeping functions acquire methylation.

## Discussion

Molecular pathways involved in epigenetic gene regulation evolve surprisingly rapidly and DNA methylation is no exception. Our work adds to the complex picture of how DNA methylation patterns change across evolutionary time and offers new insight into potential factors influencing the distribution of DNA methylation within genomes.

### Plasticity of DNA methylation landscapes

Prior to this study, DNA methylation had been characterised across insects [13] but only isolated species from more basal arthropod clades had been studied [15,23–26,30]. By examining a phylogenetically broad range of arthropod methylomes we reconstructed the trajectory of DNA methylation patterns across the phylum. Our data show that ancestral arthropods likely had moderate genome-wide methylation including methylation of a small number of transposable elements. Methylation of genes was also prominent and was enriched in exons over introns; however, the magnitude of the difference between exonic and intronic methylation was not as striking as in insects such as *A. mellifera* reflecting the presence of a higher background genomic methylation. Crucially our data also show that changes in methylation patterns can evolve rapidly within individual lineages. Most strikingly, we find strong enrichment of TE methylation evolved independently in the centipede *S. maritima* and the mealybug *P. citri*, which very likely occurred independently. This enrichment does not correlate to any obvious change in genome structure such as increased TE proportion or genome size, however it is interesting that a recent paper reported acquisition of a relatively recent TE family in *S. maritima* that acquires high levels of methylation [15], which may underpin gain of TE methylation in that species.

It is intriguing that the two species with high TE methylation had independently acquired methylation of promoters of silent genes, whilst the exons of these genes are devoid of methylation. Gene regulation by promoter methylation is also found in mammals and was likely acquired independently in the sponge *Amphimedon queenslandica* [16]. In all these cases TE methylation is also prominent so it is possible that the two are linked, perhaps relating to a requirement to control TE-derived promoter regions; however testing this hypothesis would

require experimental manipulation of methylation in *P. citri* or *S. maritima* which is currently not possible.

It is curious that repeated acquisition of similar types of DNA methylation occurs across phylogenies. This may indicate that targeting of DNA methylation to new regions can be achieved with very few genetic changes. In vertebrates, a possible example is the KRAB-Zinc finger proteins which can recruit DNA methylation to TEs through sequence-specific binding [44]. Further work to identify potential "pioneer" factors that recruit DNMTs to specific regions and underlie the divergence of methylation patterns between species will be of great interest.

## Potential factors influencing methylation of genes

Our study confirms earlier studies indicating that the most widely conserved feature of arthropod methylomes is methylation of genes, biased towards exon methylation [17]. Additionally, we confirm insights from insects and crustaceans that broadly expressed, housekeeping genes are more likely to be targeted for methylation than tissue-specific genes [26,34]. This is similar to observations in other animal groups including basal metazoans [45]. Moreover, several analyses in plants provide compelling evidence that genes with consistent expression across tissues and lifestages and evolve at a slower rate tend to have higher levels of methylation [46–49]. These facts suggest that methylation of transcribed gene bodies has functional importance in plants; however exactly what the function is unclear and still debated. For example, gene body methylation has been lost completely in the land plant *Eutrema salsugineum* [50], but studies have come to differing conclusions about whether this has a subtle effect on expression of these genes compared to orthologues in species retaining gene methylation [28,51]. Taken together with studies in animals, it does seem clear that methylation of gene bodies has an ancient evolutionary origin [48,52]. Nevertheless exactly what the function of this modification is remains to be elucidated. Similar to the case of *E. salsugineum* in plants, our data shows that within arthropods it is dispensable even in species where DNA methylation is present in other regions of the genome, such as the two crustaceans that we examined.

Whilst we cannot decipher the function of exon-enriched DNA methylation, our analyses potentially offer new insights into the molecular mechanisms whereby DNA methylation might be deposited. We identify a remarkable methylation pattern across many arthropods such that methylation levels vary periodically with the nucleosome-repeat length. This striking genome-wide pattern that we observe in some species, in particular *S. maritima*, has not been observed to our knowledge in any animal species previously. However, there are specific regions within the human genome that display apparently nucleosome length periodicity in the correlation between adjacent sites [37]; furthermore the influence of nucleosomes on methylation by DNMT3B was observed in human and yeast cells [53,54]. Moreover, DNA methylation levels show a 10bp periodicity in *Arabidopsis*, corresponding to methylation targeting nucleotides on the same face of the nucleosome [36]. Together these observations reflect a positive correlation between nucleosome occupancy and DNA methylation in *Arabidopsis* and mammals [36]. Exons are known to have better positioned nucleosomes than introns [55,56] which might explain why exons are enriched in methylation across species. We also find that promoters of genes with high levels of methylation tend to carry a clear nucleosome positioning pattern, typical of housekeeping genes, where nucleosome occupancy is high upstream and just downstream of the TSS with a nucleosome-free region between the two [42,43]. Both nucleosome positioning and DNA methylation could be linked to transcription. Since tissue-specific genes are highly expressed in only a few cell types, this might explain why they do not appear methylated in whole animal bisulphite sequencing. This would also explain

why across all species genes with very low expression are depleted of methylation (Fig 4D). Alternatively, nucleosomes themselves could dictate where DNA methylation takes place. Supporting this point there is little periodicity in DNA methylation in introns compared to exons (S8 Fig), suggesting that transcription itself is insufficient to account for this effect.

Importantly, the fact that we see these patterns based on nucleosome positioning in *Drosophila*, where DNA is not methylated, suggests that nucleosome positioning may cause differences in DNA methylation. Consistent with this idea, we show that nucleosome occupancy is high at the promoters of methylated genes that do not have housekeeping functions in *Drosophila*. Furthermore, nucleosome occupancy is low at the promoters of unmethylated housekeeping genes. Thus, we suggest that nucleosome positioning, rather than housekeeping function, may be a primary determinant of variation in DNA methylation across arthropod genomes. The observed enrichment of methylation at housekeeping genes may therefore be a consequence of the fact that housekeeping genes tend to have distinct patterns of nucleosome occupancy, and not directly related to a function for DNA methylation in regulating housekeeping genes. Such a view is supported by the fact that the effect of loss of gene body methylation on gene expression is subtle [28] and that DNMT1 knockout in the milkweed bug has little consequence for gene expression but still seems essential for embryo viability, implying that there may be distinct functions of DNA methylation unrelated to methylation of a subset of genes [34]. To test these ideas directly will require detailed examination of nucleosome positioning data across arthropods alongside knockouts of DNA methyltransferases. Nevertheless, our analyses may prompt a search for how nucleosome occupancy might determine methylation patterns across eukaryotes.

## Methods

### DNMT identification

To identify species that have retained or lost the DNA methylation pathway, we searched for homologues of DNMT. For each species, we used DIAMOND [57] to perform BLASTp searches against all annotated proteins, with *A. mellifera* DNMT1 (NM001171051), DNMT2 (XM006562945) and DNMT3 (NM001190421) as query sequences. We used InterProScan to screen out hits that lacked the C-5 cytosine-specific DNA methylase domain, and NCBI BLASTP to screen out bacterial contaminants (i.e. hits that were more similar to bacterial DNMTs than eukaryotic DNMTs). To classify DNMTs into subclades (DNMT1, 2 & 3) we aligned all homologues with MAFFT, screened out badly-aligned regions with Gblocks [58], and inferred a neighbour-joining phylogenetic tree under the Jukes-Cantor model using Geneious v10.1.3 (https://www.geneious.com).

### Genome annotation

To annotate exons in each genome we used existing annotations, excluding genes that were split across multiple contigs. To annotate regions which may contain promoters or enhancers, we took 1,000 bases upstream of each gene, excluding genes where this exceeded the contig start or end point. We annotated introns based on the position of exons, excluding genes that were split across multiple contigs (using intron_finder.py script available at https://github.com/SamuelHLewis/BStoolkit/). To annotate TEs, we used RepeatModeller v1.0.8 to generate a model of TEs for each genome separately, and then RepeatMasker v4.0.6 to annotate TEs based on the model for that genome. Within each TE, we used interproscan [59] to search for the following TE-associated domains: PF03184, PF02914, PF13358, PF03732, PF00665 & PF00077.

To annotate rRNA, we either used existing annotations or RNAmmer v1.2 [60]. To annotate tRNA, we either used existing annotations or tRNAscan-SE v1.3.1 [61]. To avoid ambiguous results caused by overlapping features, we screened out any TE annotations that overlapped any rRNA, tRNA or exon, and any upstream regions which overlapped any TE, rRNA, tRNA or exon.

## Whole genome bisulphite sequencing

To measure DNA methylation on a genome-wide scale, we carried out whole-genome bisulphite sequencing. We used the DNeasy Blood and Tissue kit (QIAGEN) according to the manufacturer's protocol to extract DNA from adult somatic tissues of the following species: *L. polyphemus*, *P. tepidariorum*, *S. maritima*, *A. vulgare*, *B. germanica*, *A. pisum*, *B. terrestris*, *N. vespilloides*, *H. melpomene* and *D. melanogaster*. For *I. scapularis*, we used the same method to extract DNA from the IDE2, IDE8 and ISE18 cell culture. To estimate bisulphite conversion efficiency, we added a spike-in of unmethylated DNA (P-1025-1, EpiGentek) equal to 0.01% of the sample DNA mass to each sample. We then prepared whole-genome bisulphite sequencing libraries from each DNA sample using the Pico Methyl-Seq Library Prep Kit (Zymo Research), according to the manufacturer's protocol (S1 Table for detailed sample metadata and sequence accession codes). We sequenced these libraries on an Illumina HiSeq 2500 instrument to generate 100bp paired-end reads. We used pre-existing whole-genome bisulphite sequencing datasets for *P. hawaiensis* (SRR3618947, [23]) and *A. mellifera* (SRR1790690, [62]).

To generate bisulphite sequencing data for *P. citri*, we extracted DNA from adult females using the DNeasy Blood and Tissue kit (QIAGEN) according to the manufacturer's protocol. To estimate bisulphite conversion efficiency, we included a spike-in of non-methylated *Escherichia coli* lambda DNA (isolated from a heat-inducible lysogenic *E. coli* W3110 strain, provided by Beijing Genomics Institute (BGI), GenBank/EMBL accession numbers J02459, M17233, M24325, V00636, X00906). Sequencing of bisulphite libraries was carried out by BGI on an Illumina HiSeq 4000 instrument to generate 150bp paired-end reads.

## Bisulphite sequencing data analysis

Before mapping reads to the genome, we trimmed sequencing adapters from each read, and then trimmed 10 bases from the 5' and 3' end of each read (using the script https://github.com/SamuelHLewis/BStoolkit/blob/master/BStrim.sh). We aligned bisulphite sequencing reads to each genome using Bismark v0.19.0 [63] in—non_directional mode with default settings. We used MethylExtract v1.9.1 [64] to estimate the level of methylation at each CpG site, calculated as the number of reads in which the cytosine is methylated divided by the total number of reads covering the cytosine, excluding sites covered by fewer than 10 reads on each strand. Due to the large number of contigs in their genome assemblies exceeding the memory limit for MethylExtract, we split the genomes of *I. scapularis*, *L. polyphemus* and *P. hawaiensis* into individual contigs, ran MethylExtract on each contig separately, and concatenated the resulting output files into one file for each genome.

To estimate the genome-wide background level of CpG methylation, we calculated the mean methylation for all CpGs outside annotated features (exon, intron, upstream region, TE, rRNA & tRNA). To gain an accurate estimate of the methylation level of each feature, we calculated the mean methylation level of all CpGs within that feature, excluding any feature with fewer than 3 sufficiently-covered CpGs (only CpGs covered by >10 reads are analysed). We estimated 95% confidence intervals for the mean methylation of genes and TEs within each species using 1000 nonparametric bootstrap replicates (i.e. genes or TEs were resampled with replacement 1000 times to generate an empirical distribution of the mean).

## Phylogenetics and ancestral state reconstruction

To infer the ancestral levels of genome-wide methylation across 29 species of arthropods with newly-produced or publicly-available methylation data (Fig 1), we obtained a time-scaled species tree from TimeTree (www.timetree.org, accessed 01.03.2020). We then used a maximum-likelihood approach to infer the genome-wide methylation level at all internal nodes of this tree based on the levels at the tips, using the fastAnc function within phytools [65].

To infer the ancestral levels of gene-body and TE methylation for the 14 focal species, we constructed a Bayesian time-scaled species tree for 14 focal species (Figs 2 & 3). We first identified 236 proteins present as 1:1:1 orthologues across our species set, concatenated the protein sequences together, and aligned them using MAFFT v7.271 [66] with default settings. We then screened out poorly-aligned regions using Gblocks [58] with least stringent settings. Using this alignment, we constructed a phylogenetic tree using BEAST v1.8.4 [67] to infer branch lengths. We specified a strict molecular clock, gamma-distributed rate variation, no invariant sites, and a birth-death speciation process. We fixed the topology and set prior distributions on key internal node dates (Arthropoda = 568 ± 29, Insecta–Crustacea = 555 ± 33, Insecta = 386 ± 27, Hymenoptera–Coleoptera–Lepidoptera–Diptera = 345 ± 27, Coleoptera–Lepidoptera–Diptera = 327 ± 26), deriving these values from an existing phylogenetic analysis of arthropods [68]. We ran the analysis for 10 million generations, and used TreeAnnotator [67] to generate a maximum clade credibility tree. We then used a maximum-likelihood approach to infer the gene-body and TE methylation levels (separately) at all internal nodes of this tree, using the fastAnc function within phytools [65].

To test whether genome-wide methylation levels differ between species with and without ALKB2, we fitted a phylogenetic mixed model using MCMCglmm [69]. To account for phylogenetic non-independence caused by sampling species with different levels of relatedness, we used the branch lengths of the time-scaled (ultrametric) species tree (see above) to calculate a genetic distance matrix, and included this in the model as a random factor. We ran the analysis for 6 million iterations, with a burn-in of 1 million iterations and thinning of 500 generations.

## RNA-Seq data analysis

To investigate the link between DNA methylation and transcription, we used RNA-Seq data generated previously for arthropod somatic tissue (NCBI PRJNA386859, [70] and the *I. scapularis* IDE-8 cell line (SRR1756347, Arthropod Cell Line RNA Seq initiative, Broad Institute, broadinstitute.org). To measure the expression of each feature, we trimmed adaptors and low-quality ends using Trim Galore with default settings, and mapped RNA-Seq reads to the genome of each species using TopHat2 v2.1.1 [71] with default settings for strand-specific libraries (—library-type fr-firststrand mode). We counted the number of reads overlapping each feature using BEDTools coverage v2.25.0 in strand-specific mode, and divided the number of reads by the feature length to generate expression level estimates in fragments per per kilobase million (FPKM).

To test whether variation in tissue-specific expression differs between highly- and lowly-methylated genes, we calculated the coefficient of variation for expression of each gene in each species with RNA-Seq data (i.e. excluding *B. germanica*, *I. scapularis* & *P. hawaiensis*). For *S. maritima* we used RNA-Seq data for fat body and nerve chord; for *P. citri* & *A. pisum* we used RNA-Seq data for female soma and germline; and for all other species we used RNA-Seq data for female and male soma and germline.

## Periodic correlation in methylation levels

To obtain an estimate of how the correlation between the methylation levels of sites varied with distance between the sites, we collected all pairs of sites separated by $d$ nucleotides where $d$ could vary between 3 and 500 nucleotides within the same exon. For each separate $d$ we then

computed the correlation coefficient across all the pairs. To quantify the periodic component of the signal we subtracted any gradual change in correlation across the entire window by calculating the residuals of a linear model. This signal was subjected to Fourier analysis using the fast Fourier transform algorithm implemented in R. A linear model was used to subtract the baseline across the 500bp and the residuals were used as a time series for input into the algorithm, with 50000 0 values ended on to the end of the series to increase the resolution of the algorithm. The total intensity of the components between 140 and 200 base pairs was calculated to give the nucleosome periodicity for each species.

## Nucleosome positioning analysis

The genomic coordinates of the *D. melanogaster* members of orthogroups conserved across all species were extracted and the top 20% (high methylation) and bottom 20% methylation (low methylation) levels selected. Nucleosome positioning data from the *D. melanogaster* S2 cell line was downloaded from Modencode [38]. The average signal was computed across 200bp windows spanning 2kb either side of the annotated transcription start site for each gene. The mean signal was computed within the high methylation and low methylation sets separately and a loess fit performed. To obtain confidence intervals, the mean signal was computed on 100 random samples containing 90% of the data and a loess fit calculated on the lowest and highest values obtained for each 200bp window.

## CAGE data analysis

Total body RNA was extracted from L3 *Drosophila melanogaster* ($w^{1118}$) larvae using the Qiagen RNeasy kit. CAGE library preparation was performed using the nAnT-iCAGE protocol [72]. Two biological replicates were prepared from 5 ug of total RNA each. The libraries were sequenced in single-end 50 bp-pair mode. CAGE tags (47 bp) were mapped to the reference *D. melanogaster* genome (assembly Release 6) using Bowtie2 [73] with default parameters. Uniquely mapped reads were imported into R (http://www.R-project.org/) as bam files using the standard workflow within the CAGEr package [74]. The 5' ends of reads are CAGE-supported transcription start sites (CTSSs) and the number of tags for each CTSS reflects expression levels. Raw tags were normalised using a referent power-law distribution and expressed as normalized tags per million (TPMs). Biological replicates were highly correlated ($r^2 = 0.99$) and were therefore merged prior to downstream analyses using standard Bioconductor packages (http://www.bioconductor.org/) and custom scripts.

CTSSs were clustered together into tag clusters, a single functional transcriptional unit, using distance-based clustering, with the maximum distance allowed between adjacent CTSSs being 20 bp. For each tag cluster, the interquantile width was calculated as the distance between CTSSs at the 10th and 90th quartile of the cumulative distribution of expression across the cluster. The interquartile range of each gene within the top 20% and bottom 20% of methylation levels was extracted and compared.

## Supporting information

**S1 Fig. ALKB2 DNA repair is associated with high levels of DNA methylation across arthropods.** Boxplot showing genome-wide methylation levels in 31 arthropod species with and without ALKB2.
(PDF)

**S2 Fig. Metagene plot of methylation within TEs and in flanking sequence for all species.**
TEs are shown in pink, flanking sequence in white.
(PDF)

**S3 Fig. Domain structure of DNMT1s from arthropods, with the human DNMT1 for comparison.**
(AI)

**S4 Fig. Alignment of the CXXC domain from DNMT1 highlighting the disruption of this domain in *P. citri*.**
(AI)

**S5 Fig. Methylation of single exon and multi-exon genes for all species in which we see gene body methylation.**
(PDF)

**S6 Fig. Expression patterns of methylated and unmethylated genes for all species (cf Fig 5B).**
(PDF)

**S7 Fig. Fourier transform analysis of periodicity in DNA methylation correlation.** A) Diagram of method to convert from the pattern into the magnitude of the nucleosome periodicity B, C) Periodicity in coding sequences and transposable elements respectively.
(PDF)

**S8 Fig. Intron periodicity is markedly less apparent than exon periodicity.** *S. maritima* exons 1 to 4 (A) and introns 1 to 4 (B) are shown for comparison.
(PDF)

**S9 Fig. Methylation is a better predictor of nucleosomal pattern than housekeeping gene status.** Both housekeeping genes and non-housekeeping genes that are methylated genes show enhanced nucleosome occupancy at the +1 nucleosome.
(PDF)

**S1 Table. Details of the tissue type, sex, caste, BioSample Accession and SRA Accession of each sample that was newly-sequenced in this study.**
(XLSX)

## Acknowledgments

We thank L. Bell-Sakyi, the Tick Cell Biobank, A. McGregor, R. Jenner, M. Akam, A. McLean, D. Collins, R. Kilner, A. Pinharanda and C. Jiggins for providing arthropod samples, and G. Filion for providing a curated list of *D. melanogaster* housekeeping genes. PS is a recipient of an EMBO Young Investigator award. Sequencing of bisulphite and CAGE libraries was carried out by the LMS Genomics Facility.

## Author Contributions

**Conceptualization:** Samuel H. Lewis, Eric A. Miska, Francis M. Jiggins, Peter Sarkies.

**Data curation:** Samuel H. Lewis.

**Formal analysis:** Samuel H. Lewis, Francis M. Jiggins, Peter Sarkies.

**Funding acquisition:** Eric A. Miska, Francis M. Jiggins, Peter Sarkies.

**Investigation:** Samuel H. Lewis, Laura Ross, Stevie A. Bain.

**Resources:** Laura Ross, Eleni Pahita, Stephen A. Smith, Richard Cordaux, Eric A. Miska, Boris Lenhard.

**Supervision:** Laura Ross, Boris Lenhard, Francis M. Jiggins, Peter Sarkies.

**Writing – original draft:** Francis M. Jiggins, Peter Sarkies.

**Writing – review & editing:** Samuel H. Lewis, Laura Ross, Boris Lenhard, Francis M. Jiggins, Peter Sarkies.

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
