## [Decision Letter · Decision Letter 0]

4 Feb 2020

Dear Peter

Thank you very much for submitting your Research Article entitled '­­­­­­Widespread conservation and lineage-specific diversification of genome-wide DNA methylation patterns across arthropods' to PLOS Genetics. Your manuscript was fully evaluated at the editorial level and by independent peer reviewers. The reviewers appreciated the attention to an important problem, but raised some substantial concerns about the current manuscript. Based on the reviews, we will not be able to accept this version of the manuscript, but we would be willing to review again a much-revised version. We cannot, of course, promise publication at that time.

If you decide to revise the manuscript for further consideration at PLOS Genetics, please aim to resubmit within the next 60 days, unless it will take extra time to address the concerns of the reviewers, in which case we would appreciate an expected resubmission date by email to plosgenetics@plos.org.

[LINK]

We are sorry that we cannot be more positive about your manuscript at this stage. Please do not hesitate to contact us if you have any concerns or questions.

Yours sincerely,

Wolf Reik

Section Editor: Epigenetics

PLOS Genetics

Wolf Reik

Section Editor: Epigenetics

PLOS Genetics

Reviewer's Responses to Questions

**Comments to the Authors:**

Reviewer #1: The study be Lewis and colleagues’ profiles genome-wide DNA methylation at single base resolution in species distributed broadly across the arthropod phylum. The observed repeated loss of DNA methylation and its associated DNMTs in some species indicates that the most recent common ancestor possessed DNA methylation. There is widespread evidence that DNMT3 is frequently lost across this phylum, yet, DNMT1 is sufficient to maintain the presence of CpG methylation in the genome. In a couple of species they observed independent acquisition of promoter methylation which was associated with TE acquiring. This is a very exciting observation. Curiously, DNA methylation in arthropods is highly enriched in gene bodies, especially within exons. The authors observed a periodicity associated with DNA methylation that very likely reflects the distribution of nucleosomes. To test this possibility, the authors took advantage of the fact that genes with methylation in arthropods are generally conserved. This provided an opportunity to use nucleosome positioning data from Drosophila, a species without DNA methylation. The authors found that nucleosomes were much better positioned in genes that are generally gene body methylated across the arthropod phylum.

Overall, this is a very nice and timely study that contributes to our growing knowledge that although the DNA methylation machinery is generally conserved, the manner in which the genome uses it are quite diverse. I find it very exciting that the authors found evidence for independent acquisition of promoter methylation that can influence gene expression. These unique species will provide clues on how this process has evolved. The minor comments below are intended to improve upon this nice contribution to the field.

1. I could not find any summary sequencing statistics associated with the WGBS data. Number of aligned reads, coverage, sodium bisulfite conversion efficiencies, etc.

2. Although I really like the use of the comparative analysis using Drosophila nucleosome positioning data, I think the manuscript should be carefully modified to indicate that there is not yet direct data of nucleosome positioning in a species that possesses DNA methylation (at least from this study). I believe the results will be the same, but until the experiment is done the language should be carefully crafted to make this clear.

3. In the species that independently acquire DNA methylation do the DNA methyltransferase acquire additional domains that could influence its ability to target different regions of the genome? Please add this to the discussion.

4. I would also consider that in addition to targeting methylation to exons an alternative explanation that DNA methylation maintenance might be more efficient at these regions. Targeting and maintenance are a bit different and these possibilities could be discussing.

5. In the first figure many methylomes are of low coverage, but a high coverage methylation that was not included is from Oncopeltus fasciatus.

Reviewer #2: Sarkies et al. analyze DNA methylation patterns in a range of arthropod species. They find that most arthropods have low levels of TE methylation, as is common in invertebrates, but that two species have preferential TE methylation that likely evolved independently. This is especially convincing in S. maritima. The authors also provide evidence that promoter methylation causes transcriptional silencing in these species and suggest that this is linked to methylation as a mechanism of TE silencing. A recent paper from Ryan Lister’s lab argues this for a sponge (de Mendoza et al, Nat Ecol Evol. 2019) and echoes a much earlier proposal for the evolution of TE silencing by methylation in vertebrates (Zemach et al., Science 2010). Still, it’s valuable to have additional examples that demonstrate the ease with which this evolutionary development can occur in animals.

The authors’ treatment of gene body methylation is of more concern. The association of gene body methylation with constitutively expressed, evolutionarily conserved housekeeping genes and its corresponding depletion from variably expressed genes is probably the best-known gene body methylation characteristic in plants and animals. The initial description of this phenomenon was published by Steve Jacobsen’s lab in Arabidopsis (Zhang et al., Cell 2006); more elaborate characterizations in Arabidopsis and honeybee were published a little later (Aceituno et al., 2008; Elango et al., 2009). Many more papers have been published since, as illustrated by this quote from a recent paper that reported this phenomenon in the coral Acropora millepora (Dixon et al., Mol Biol Evol 2016):

“We showed that strongly methylated genes in A. millepora tend to have constitutive and ubiquitous functions and are less likely to be differentially expressed across developmental stages and environmental regimes. These results corroborate earlier findings from diverse taxa including plants (Aceituno et al. 2008; Coleman-Derr and Zilberman 2012; Takuno and Gaut 2012), cnidarians (Sarda et al. 2012; Dimond and Roberts 2016), mollusk (Gavery and Roberts 2010, 2013), arthropods (Elango et al. 2009; Wang et al. 2013), and a basal chordate (Suzuki et al. 2013; Keller et al. 2015). The relationship with differential expression in response to environmental regimes suggests the intriguing possibility that gbM could modulate gene expression plasticity.”

Since then, this phenomenon has been reported in several more papers (probably not a complete list): Kvist et al., Genome Biol Evol 2018 for Daphnia; Gatzmann et al., Epigenetics & Chromatin 2018 for crayfish; Liu et al., Genes 2019 for spiders; Harris et al., Epigenetics & Chromatin 2019 for honeybee. Note that all these papers cover arthropods (and half are not cited). The crayfish paper in particular may complicate the authors’ conclusions about crustaceans.

Therefore, it is not “remarkabl[e]” that “the same set of genes are likely to be methylated in all species” or that “these genes have characteristic patterns of expression”. This has been extensively demonstrated in many plant and animal (including arthropod) species. The authors are not confirming an “earlier speculation” as they put it in the discussion, but restating well-supported conclusions from earlier work.

There are similar issues about the association between genic DNA methylation and nucleosomes. First, the authors’ claim that “nucleosome positioning influences DNA methylation levels across arthropods” (line 3, page 10) is far stronger than the data justify. Instead, the authors report a correlation with nucleosome positioning in a species (Drosophila) that lacks DNA methylation. They can’t distinguish if nucleosome positioning influences methylation, vice versa, or both. Second, the relationship between methylation and nucleosomes is hardly unexplored. Of most relevance, enrichment of methylation over nucleosomes and the corresponding nucleosomal periodicity of methylation have been reported long ago for Arabidopsis and humans (Chodavarapu et al., Nature 2010). An enrichment of gene body methylation at nucleosomes has been described more recently in Arabidopsis (Lyons and Zilberman, eLife 2017). Preferential methylation of nucleosomes in genes is not a new finding. Other strong associations between nucleosomes and DNA methylation (cytosine and adenine) have also been reported, including the ability of methylation to influence nucleosomes (for example, Huff and Zilberman, Cell 2014; Fu et al., Cell 2015; Beh et al., Cell 2019). Finally, the authors are creating the appearance of novelty by associating known features of housekeeping gene transcription and nucleosome organization with a known feature of gene body methylation (enrichment in constitutively expressed housekeeping genes). Because gene body methylation is concentrated in housekeeping genes, it will of necessity correlate with features of these genes.

This paper contains interesting observations about the evolution of DNA methylation and could be a valuable addition to the field if the authors were to place their finding properly within the existing literature. However, given the relatively low novelty of the conclusions, I feel this paper is much more suitable for a journal like Mol Biol Evol or Epigenetics & Chromatin (in which many of the relevant recent papers have been published).

Reviewer #3: DNA methylation (5MeC) is an epigenetic regulatory mechanism, which among eukaryotes displays great variability in terms of genomic content and function. The majority of studies focusing on 5MeC to date have been carried out on vertebrate model organisms that exhibit hypermethylated genomes. Due to the recent increase in accessibility of genomic sequencing technologies, more invertebrate organisms have had their genomes and DNA methylomes sequenced. While this provided important insights into the evolutionary conservation and divergence of 5MeC, many open questions still remain. In the current manuscript Lewis et al generate base-resolution DNA methylome maps of the highly diverse Arthropod phylum. While it is fair to say that none of the reported findings come across as particularly surprising, this is an important piece of work that will help in better understanding the evolution of 5MeC. This work will also provide a useful genomic resource that will enable further evolutionary studies in the 5MeC field. My comments for improvement can be found below.

1) The authors state that the sequence data is available via SRA. However, the PRJNA589724 accession number does not work. I could also not find any reviewer links that would point to these data. This needs to be fixed before publication.

2) The samples are poorly described. It is not clear what type of tissue / organ / cell type / developmental stage was used for DNA extraction / library preps.

3) P6, L24-25 and P7, L6-7: Targeting of TEs by 5MeC was previously shown in S maritima. A citation is required here (de Mendoza et al, 2019, Genome Res).

4) Figure 3C. Could the authors speculate on the differences in 5MeC exon enrichment directionality? For example Limulus, Exodes, Parasteatoda show more enrichment at 5' ends whereas Bombus and Apis, display more enrichment at 3' gene ends.

5) Would some of the published RNA-seq data allow for analyses of the frequency of cryptic transcription initiation in organisms with low gene body 5MeC levels vs the ones with robustly methylated GBs?

6) Related to Figs, 4,5,6: It is not clear from which tissues / organs / cell types / developmental stages RNA-seq data originates from and how this compares to the DNA methylome data. This needs to be clearly explained. Could these discrepancies be the reason for the reversed pattern in P citri or the observation that many highly conserved and expressed genes lack 5MeC?

**Have all data underlying the figures and results presented in the manuscript been provided?**

Reviewer #1: No: Missing sequencing summary statistics

Reviewer #2: None

Reviewer #3: Yes

PLOS authors have the option to publish the peer review history of their article (what does this mean?). If published, this will include your full peer review and any attached files.

Reviewer #1: No

Reviewer #2: No

Reviewer #3: No

---

## [Decision Letter · Decision Letter 1]

3 May 2020

Dear Peter,

Thank you very much for submitting your Research Article entitled '­­­­­­Widespread conservation and lineage-specific diversification of genome-wide DNA methylation patterns across arthropods' to PLOS Genetics. Your manuscript was fully evaluated at the editorial level and by independent peer reviewers. The reviewers appreciated the attention to an important topic but identified some aspects of the manuscript that should be improved.

We therefore ask you to modify the manuscript according to the review recommendations before we can consider your manuscript for acceptance. Your revisions should address the specific points made by each reviewer.

[LINK]

Yours sincerely,

Wolf Reik

Section Editor: Epigenetics

PLOS Genetics

Reviewer's Responses to Questions

**Comments to the Authors:**

Reviewer #1: The revisions to this manuscript are well done. This is going to make a nice contribution to the field.

Reviewer #2: I am grateful to the authors for seriously considering my concerns and revising their manuscript accordingly. The paper now far more accurately reflects the existing state of knowledge and the authors’ new contributions. I remain sceptical about the appropriateness of this manuscript for PLoS Genetics, but this is ultimately a matter for the editors.

I do have one suggestion for improvement. The conservation of gene body methylation across orthologs has been extensively studied in plants (Takuno and Gaut, PNAS 2013; Seymour et al., PLoS Genetics 2014; Takuno et al., MBE 2017; Seymour and Gaut, MBE 2019). This work reaches the same conclusion the authors do here, that the same sets of genes tend to be methylation across species (spanning diverse timescales). Furthermore, genes that retain methylation across evolution have a higher tendency to have conserved sequences and to exhibit stable and constitutive expression (a rough proxy for housekeeping functions) than genes that have species-specific methylation. This work should be cited and discussed. I also suggest the authors test if genes that retain gbM across arthropods also have a higher tendency for conservation and stable expression, and perhaps a specific nucleosome configuration. This would make a valuable addition to the paper.

**Have all data underlying the figures and results presented in the manuscript been provided?**

Reviewer #1: Yes

Reviewer #2: None

PLOS authors have the option to publish the peer review history of their article (what does this mean?). If published, this will include your full peer review and any attached files.

Reviewer #1: No

Reviewer #2: No

---

## [Editor Report · Decision Letter 2]

15 May 2020

Dear Peter,

We are pleased to inform you that your manuscript entitled "­­­­­­Widespread conservation and lineage-specific diversification of genome-wide DNA methylation patterns across arthropods" has been editorially accepted for publication in PLOS Genetics. Congratulations!

Yours sincerely,

Wolf Reik

Section Editor: Epigenetics

PLOS Genetics

Comments from the reviewers (if applicable):

**Data Deposition**

http://datadryad.org/submit?journalID=pgenetics&manu=PGENETICS-D-19-02110R2

**Press Queries**

---

## [Editor Report · Acceptance letter]

17 Jun 2020

PGENETICS-D-19-02110R2 

­­­­­­Widespread conservation and lineage-specific diversification of genome-wide DNA methylation patterns across arthropods 

Dear Dr Sarkies, 

We are pleased to inform you that your manuscript entitled "­­­­­­Widespread conservation and lineage-specific diversification of genome-wide DNA methylation patterns across arthropods" has been formally accepted for publication in PLOS Genetics! Your manuscript is now with our production department and you will be notified of the publication date in due course.

With kind regards,

Matt Lyles

PLOS Genetics

On behalf of:
